# Excessive Noise in Neonatal Units and the Occupational Stress Experienced by Healthcare Professionals: An Assessment of Burnout and Measurement of Cortisol Levels

**DOI:** 10.3390/healthcare11142002

**Published:** 2023-07-12

**Authors:** Jocélia Maria de Azevedo Bringel, Isabel Abreu, Maria-Cláudia Mendes Caminha Muniz, Paulo César de Almeida, Maria-Raquel G. Silva

**Affiliations:** 1Faculty of Science and Technology, University Fernando Pessoa, 4249-004 Porto, Portugal; 2FP-I3ID, University Fernando Pessoa, 4249-004 Porto, Portugal; 3Postgraduate Program in Neuropsychology, Centro Universitário Christus, Fortaleza 60160-230, Brazil; fgaclaudia10@gmail.com; 4Postgraduate Program in Clinical Health Care Nursing, Universidade Estadual do Ceará, Fortaleza 60714-903, Brazil; paulo.almeida@uece.br; 5Faculty of Health Sciences, University Fernando Pessoa, 4200-150 Porto, Portugal; 6CIAS-Research Centre for Anthropology and Health—Human Biology, Health and Society, University of Coimbra, 3000-456 Coimbra, Portugal; 7CHRC-Comprehensive Health Research Centre, Nova Medical School, Nova University of Lisbon, 1150-090 Lisbon, Portugal

**Keywords:** noise, sound level pressure, work-related stress, Burnout Syndrome, neonatal intensive care, cortisol

## Abstract

Excessive noise in the work environment has been associated with extra-auditory symptoms, which can have harmful long-term effects on individuals. The purpose of this study was to identify noise levels in neonatal intensive care units and investigate their impact on the occurrence of stress among healthcare professionals, using cortisol levels as a biomarker for Burnout Syndrome. This descriptive, observational, and cross-sectional study was conducted in four public teaching hospitals in Fortaleza, Ceará, Brazil. Sound pressure levels in the environment were measured, and questionnaires were administered to collect sociodemographic data and assess perceptions of the work environment and Burnout symptoms. Saliva samples were collected at the beginning and end of work shifts for cortisol quantification. The average sound pressure ranged from 59.9 to 66.4 dB(A), exceeding the recommended levels set by Brazilian and international legislation. Among the 256 participants, the average age was 39.4 years, with 95% being female. The majority (70.9%) were nurses, and 22.7% were physicians. There was no significant association found between noise and Burnout Syndrome, nor with changes in cortisol levels. However, a significant association was observed between the perception of excessive noise and the sensation of a stressful work shift (*p* = 0.012). All evaluated professionals displayed symptoms of Burnout. The high sound pressure levels indicated that the assessed environments did not meet the recommended standards for acoustic comfort, and this was associated with the participants’ perception of stressful work shifts. While Burnout symptoms were evident in our participants, it was not possible to confirm a correlation with high noise levels.

## 1. Introduction

The context of a Neonatal Intensive Care Unit (NICU) is complex due to the emergent nature of healthcare professionals’ actions in saving the lives of newborn patients [1,2]. In addition to the specificities of patients in an immature stage of growth and development, which require utmost care and attention, there is increasing pressure and strict supervision on the professionals involved, which have been associated with stress factors [3]. Moreover, excessive workload, high demands, and responsibilities [4], long working hours, the pressure of caring for critically ill or end-of-life patients, conflicts between professionals and managers, salary dissatisfaction, and inadequate work environment are also factors for workplace stress [5].

The use of warning equipment, such as alarms and other signals, to ensure the survival of patients and the smooth operation of the care unit, often contributes to excessive environmental noise [6].

Recommendations for noise levels in hospital environments vary based on the time of day and night. Table 1 presents the values recommended by the American Academy of Pediatrics (AAP) [7], Brazilian standards [8], and the World Health Organization (WHO) [2,9]. Noise levels exceeding these recommendations do not provide acoustic comfort and can pose physical and psychological risks, leading to illness.

The effects of noise on health professionals range from complaints related to discomfort and tinnitus [10] to the occurrence of hypertension and myocardial infarction, anxiety and depression [11], suicidal ideation, and work stress [12], potentially leading to Burnout Syndrome [13].

Exposure to excessive environmental noise induces mental stress, activating the hypothalamic–pituitary–adrenal axis and resulting in cortisol release. These effects can disrupt the production of cortisol, a glucocorticoid produced and released by the adrenal gland. This, in turn, triggers an inflammatory process with detrimental effects on various systems, including the immune system [14].

Salivary cortisol quantification serves as a reliable biological marker in stressful situations [15] and is applied in many studies. Bauer et al. (2000) found that elderly caregivers of chronically ill individuals experienced greater distress and in-creased basal salivary cortisol compared to non-caregivers [16]. Similarly, Rojas-González et al. (2004) observed that professionals in the brewing industry exposed to noise exhibited elevated cortisol levels at the end of their work shifts and reported extra-auditory symptoms such as headaches, insomnia, and arterial hypertension [17]. Studies involving health professionals working in intensive care units exposed to high noise levels have consistently reported experiencing mental and physical disturbances related to their work environment, with negative consequences for their health [4,10,14,18,19].

The literature has shown the incidence of Burnout Syndrome in health professionals, its causes, and consequences for health, as well as its association with shift work, chronotype, stress and work. However, there is a gap when we seek information on the occurrence of Burnout and stress in health professionals working in neonatal intensive care units (NICU), especially in an environment with high noise levels. Knowing this environment, the following questions came to us: would neonatal intensive care professionals be exposed to an adequate level of noise? Would there be a correlation between noise levels and work-related stress? The initial conception was that environments with high noise levels would be related to the increased stress of the professionals.

The aim of this study was to identify noise levels in the NICU and investigate their impact on the occupational stress of health professionals using salivary cortisol as a biomarker. The specific objectives were to identify the noise levels of the environments and the profile, chronotype and occurrence of Burnout Syndrome in the sample.

## 2. Materials and Methods

### 2.1. Subjects

A total of 256 health professionals working in four neonatal intensive care units in the district of Fortaleza, Ceará, Brazil, were evaluated between June 2019 and November 2020.

After ethical approval of the project, the initial step of our research involved contacting the head of each NICU and obtaining their respective consent to conduct the study. Following their approval, all healthcare professionals working in the units were invited to participate in a meeting, where we explained the project objectives and methodology.

Those who voluntarily agreed to participate in the study were then briefed on the procedures for collecting salivary cortisol samples and completing the questionnaire. The sample selection process adhered to specific criteria, including a requirement to spend at least six consecutive hours in the environment, willingness to participate voluntarily by signing the consent form, and a commitment to provide reliable responses on the questionnaire and collect the biological material (saliva).

Subsequently, a suitable period for noise measurements was determined. The collections were conducted on consecutive days to encompass all work shifts and involve the maximum number of professionals. This entailed four or five days of data collection in each hospital unit from Monday to Friday, ensuring representation across different days of the week.

All professionals present during the work shift in which the noise measurements were conducted were invited to participate.

The inclusion criteria required participants to be currently employed in the NICU for a minimum of 6 months and present at work for at least 6 consecutive hours on the day of cortisol collection.

Exclusion criteria involved having an incomplete questionnaire and taking corticosteroids.

All participants provided informed consent to participate in the study. The informed consent was obtained through a written form and signed individually by each participant and was stored under the custody of the researchers.

The research protocol was approved by the Ethical Committee of Plataforma Brazil under protocol number 3.158.600.

### 2.2. Procedures

Participants responded to an in-person questionnaire prepared by the authors that assessed the following data: sociodemographic, work conditions, anthropometric indicators, and self-perception of Burnout. In addition, salivary cortisol of the participants and environmental noise measurements were assessed.

#### 2.2.1. Sociodemographic and Work Conditions

A structured questionnaire prepared by the authors and subdivided into 3 parts was applied. The first part consisted of 11 questions that evaluated the following: sociodemographic data: age, sex, marital status, education, and household data (children); professional data: hospital, function, working time, shifts, other jobs, transport time, occurrences and health data: body mass index, smoking and alcohol consumption, physical examination, diseases, medication, perception of tiredness and stress, blood pressure, heart rate and salivary cortisol at the beginning and end of the work shift.

The second part of the questionnaire consisted of 7 questions that evaluated working conditions and health. Among them, work period, if there were other jobs, if yes, how many consecutive hours of work, presence of disease, and use of medication and exercise, sleep duration, chronotype and stress.

The third part of the questionnaire consisted of open questions completed by the researcher such as blood pressure, heart rate, self-reported anthropometric indicators including weight and height, perception about the participant’s physical state and time of collection of salivary cortisol at the beginning and end of the shift and intercurrences in the shift, perception about the work shift in relation to stress and fatigue, noise intensity, and front reactions when in noisy environments.

#### 2.2.2. Self-Perception of Burnout

The Brazilian version of the Burnout Characterization Scale [20] was used to evaluate the subjective stress experienced by health professionals. The scale exhibited good internal consistency, with a Cronbach’s alpha coefficient exceeding 0.70. It comprises 20 questions, and participants rate their responses on a 5-point scale (1—Never, 2—Annually, 3—Monthly, 4—Weekly, 5—Daily). The scores from the answers are totaled, leading to the following categories: 0 to 20 points (“No signs of Burnout”), 21 to 40 points (“Possibility of developing Burnout”), 41 to 60 points (“Initial phase”), 61 to 80 points (“Installed”), and 81 to 100 points (“Considerable phase of Burnout”) [21]. Additionally, four supplementary questions were included to assess the health professionals’ perception of their physical state at the beginning and end of the work shift.

#### 2.2.3. Salivary Cortisol

Salivary cortisol samples were collected from the participants before and at the end of their work shift using Salivette^®^ tubes with a synthetic fiber roll [22]. The chemiluminescence method was employed for cortisol analysis due to its reliability and precision. The results were reported in micrograms per deciliter (µg/dL) [23]. Normal salivary cortisol values were determined based on published literature, with values below 0.736 µg/dL between 6 a.m. and 10 a.m. and below 0.252 µg/dL between 4 p.m. and 8 p.m. [24]. Prior to the sample collection, participants were instructed not to consume alcohol or smoke, and to observe a 2 h interval between the collection and food intake or teeth brushing. The first sample, collected at the beginning of the work shift, was supervised by the researcher who provided instructions for the correct collection and proper storage in a refrigerated environment. The second sample, collected at the end of the work shift, was self-collected by the participant following the guidelines provided during the first sample collection, and it was also stored in a refrigerated location with appropriate identification.

#### 2.2.4. Environmental Noise Measurements

Sound pressure levels (SPL) were measured using a HIGHMED model HM-851 sound meter, which was calibrated and configured in the slow response circuit (slow) and compensation circuit A, as recommended by the Brazilian Standard [8], used as a parameter for evaluating noise in indoor environments. The recommended noise values for the hospital environment are shown in Table 1.

This device served as a parameter for evaluating noise in indoor environments. The sound meter was positioned at a distance of 100 cm from the ceiling and connected to a computer. It measured the environmental noise level every second and recorded the noise wave in a graph using the SoundLab program, version 1.0.0.18. The measurements were conducted continuously for a period of 24 h, covering 4 or 5 days during the week to ensure a comprehensive assessment of all work shifts. The work shifts were categorized as morning (7 a.m. to 1 p.m.), afternoon (1 p.m. to 7 p.m.), daytime (7 a.m. to 7 p.m.), and night-time (7 p.m. to 7 a.m.), according to the staff schedule. Throughout the measurement period, the researchers noted down the prominent noise sources in the environment.

The mean noise level (*LAeq*) was calculated using a logarithmic equation in accordance with the Brazilian Standard [8] and expressed in decibels.
$$LAeq,T=10\times \log_{10}\left[\frac{1}{n}\times \left(10^{\frac{LAeq,1s,m1}{10}}+10^{\frac{LAeq,1s,m2}{10}}+\cdots +10^{\frac{LAeq,1s,mn}{10}}\right)\right]$$
where

-*LAeq,T* is the A-weighted equivalent continuous sound pressure level integrated over a time *T* at the point evaluated;-*T* represents the total time evaluated in seconds;-*m* represents each measurement performed per second (*LAeq*,1*s*);-*n* is the total number of measurements.

#### 2.2.5. Statistical Analysis

Statistical analysis was performed using SPSS software for Macintosh, version 23 (IBM Corp., Armonk, NY, USA). For comparisons between two groups, the Student’s t-test or the Mann–Whitney test was used for normally distributed and non-normally distributed data, respectively. In the comparisons involving three groups, the ANOVA test with Tukey’s post hoc test or the Kruskal–Wallis test with Dunn’s post hoc test was used for normally distributed and non-normally distributed data, respectively. Categorical variables were presented as absolute counts and percentages and compared using the Chi-square test or Fisher’s exact test. Normality of quantitative variables was assessed using the Shapiro–Wilk test, and data asymmetry was evaluated through standard deviation, histogram analysis, and QQ plots. Normally distributed data were expressed as mean and standard deviation, while non-normally distributed data were presented as median and interquartile range (IQR). Statistical significance was set at *p* < 0.05.

#### 2.2.6. Ethical Approval

The study received ethical approval from the Ethical Committee of Plataforma Brazil and the local hospitals’ Ethical Committees under protocol number 3.158.600.

## 3. Results

### 3.1. Subjects’ Sociodemographic Characteristics

A total of 256 professionals participated in the study. The participants had a mean age of 39.4 ± 9.8 years, with the majority being female (94.9%). About 51.8% of the participants were married, and 64.9% had financially dependent children. Nursing was the most common profession, accounting for 70.9% of the participants, with 51.8% being nursing technicians and 19.1% being nurses. Additionally, 22.7% were physicians, and 6.4% were physiotherapists and speech-language pathologists.

In terms of education level, 42% of the professionals had postgraduate degrees, 26.8% had completed high school, and 31.2% had completed technical school. All professionals had been working in the field for more than 6 months, with the majority (41.8%) having less than 5 years of experience.

### 3.2. Health Professionals’ Work Conditions, Salivary Cortisol and Subjective Stress

In terms of work schedules, 42.6% of the participants worked during twelve consecutive hours, specifically the daytime shift from 7 a.m. to 7 p.m., while 34.7% worked the night-time shift from 7 p.m. to 7 a.m. In total, 22.7% of health professionals worked in shorter shifts of six hours, with 13.7% in the morning shift and 9% in the afternoon shift.

The majority of participants (80.4%) had a travel time to work of less than 60 minutes. While 63.2% of participants mentioned having only one job, 15.9% also worked at another hospital. Regarding the duration of work, 53% of professionals worked for more than 12 consecutive hours, 22.2% worked for 18 h, and 30.6% worked for 24 h.

Table 2 provides additional information on work-related aspects and characteristics associated with stress among the evaluated health professionals.

The study found that a significant portion of the participants experienced tiredness and stress during their work shifts. At the beginning of the shift, 66.5% of professionals reported feeling “a little tired” (53.5%) or “very tired” (10.3%), while only 36.2% started their shift feeling “rested.” At the end of the shift, 45.4% reported being “a little tired,” 35.7% felt “very tired,” and 18.9% felt “rested.”

About 35.8% of the professionals reported experiencing events or situations that could lead to stress. The most common stressful events were the admission of severe patients in the NICU, which required more attention and increased workload (40%), followed by acute situations such as cardiac arrest (18.6%), admission of critically ill patients (15.7%), situations resulting in death (14.3%), and clinical deterioration requiring ventilatory support and intubation (11.4%).

The health professionals’ perception of their work shift varied, with 56.6% considering it tiring, 45.7% finding it stressful, and 78.4% perceiving it as excessively noisy.

The occurrence of Burnout Syndrome was identified in 73.9% of professionals, with 15.0% classified as having the syndrome “installed” and 58.9% at the “initial phase.” None of the health professionals showed “no symptoms” of Burnout, and 26.1% exhibited signs indicating a “possibility” of developing the syndrome (Table 2).

Most health professionals in the study demonstrated adequate cortisol levels at both the beginning (80.4%) and end of their work shifts (96.5%). The increase in the frequency of adequate cortisol levels from the beginning to the end of the shift was statistically significant (*p* < 0.001). Surprisingly, there was a higher number of professionals with increased salivary cortisol (19.6%) at the beginning of the shift compared to the end of the shift (3.5%) (*p* < 0.001). It was also observed that professionals who came from another job had elevated cortisol levels.

Among the 44 professionals who showed increased cortisol at the beginning of the work shift, 18 (41.9%) were married, 14 (32.6%) were single, 9 (20.9%) were in stable relationships, and 2 (4.7%) were divorced. Interestingly, the salivary cortisol levels at the beginning of the work shift were associated with the professionals’ “marital status” (*p* = 0.010) and “coming to work from another job” (*p* = 0.020) (Table 3).

### 3.3. Environmental Noise Measurements

The sound pressure level (SPL) in the units ranged from 59.9 to 66.4 dB(A). All evaluated units showed high noise levels. A one-way ANOVA showed that the L_Aeq_ level was different for the different work shifts groups (*p* = 0.005). Tukey post hoc analysis revealed that the increase in L_Aeq_ level from the night to afternoon shift was statistically significant, but no other group differences were statistically significant (Figure 1, Table 4).

Dashed lines represent the measures recommended by Brazilian legislation [8] and by the American Academy of Pediatrics. Bars represent the means and error bars the standard deviation. 

### 3.4. Factors Associated with Exposure to Noise Levels and Stress

Taking into consideration the primary sources of noise in the NICU under investigation (equipment alarms such as infusion pumps, vital signs monitors, and heated cradles, as well as professionals’ voices), only the subjects’ work shift demonstrated a significant association with the generated noise (Table 4). While health professionals displayed a higher prevalence of normal salivary cortisol levels compared to altered levels at the start or conclusion of their work shifts, no significant correlations were observed with the noise levels in the NICU. Additionally, 78.4% of the subjects reported excessive noise.

A chi-square test revealed a significant association between the perception of “excessive noise” and the perception of a “stressful shift” (*p* = 0.012), indicating that noise is a relevant factor in increasing stress among health professionals, as demonstrated in the present study (Figure 2).

Among those who marked the shift as a “stressful shift,” it was observed that 85% also reported it as “excessive noise.” On the other hand, for those who marked the shift as a “non-stressful shift,” a lower percentage, 69%, reported “excessive noise (Figure 2).

In order to conduct a more detailed evaluation of the potential relationship between stress and noise levels, we assessed the correlation between L_Aeq_ levels (equivalent continuous sound pressure levels) and cortisol levels in various scenarios that could influence physiological cortisol levels.

Table 5 presents the correlation between cortisol levels and the noise levels participants were exposed to, while considering the occurrence of Burnout and gender as separate factors.

No significant correlations were found between the noise levels and cortisol levels when analyzing the data separately by gender or by the rating of the Burnout Syndrome. Even after adjusting the correlations for the presence of Burnout, no significant association was observed in the group with established Burnout between cortisol levels and noise levels (*p* = 0.864).

## 4. Discussion

This study revealed elevated levels of noise in the investigated Brazilian NICU, consistent with findings from other published literature [19,25,26,27,28,29,30,31,32,33].

These findings are concerning, as the recorded noise levels exceeded both Brazilian [8] and international recommendations [7] for hospitals. These results suggest that the NICU environment may potentially pose harm to both patients [34,35] and health professionals [36,37,38].

The noise generated in these intensive care units is a result of various healthcare activities involving newborns [39], such as the aspiration of secretions, the use of artificial ventilators for oxygen supply [40]; the architectural design and construction materials of the [39,41]; the distribution of beds [27,28] and especially, the inappropriate use of multi-parameter monitors and infusion pumps equipped with alarms, often disregarded by healthcare professionals, resulting in prolonged noise in the environment [37,42,43].

In our study, equipment alarms and professionals’ voices were identified as the major sources of noise, which is consistent with findings reported in the literature [27,28,35,37,39,44]. The sound produced by health professionals’ voices is frequently mentioned as a contributing factor to noise in these environments. Clinical case discussions, medical visits by specialists, bedside shift changes [29,45] and parallel conversations between professionals are also documented by several authors in the literature [29,33,46,47].

Furthermore, the presence of newborns’ parents, who are allowed to stay in the NICU, contributes to increased noise levels, but, encouraging conversations between parents and their newborns is beneficial for family bonding and the development of the newborn [39,44,48,49].

In contrast to other studies that have reported the morning shift as the loudest due to the concentration of activities and movement of professionals and students, our study found that the afternoon shift had higher noise levels, although it was also carried out in teaching hospitals [29,50].

Our findings are consistent with a study conducted in a Spanish neonatal intensive care unit (NICU), which assessed noise levels over a 20-day period. The Spanish study found that noise consistently remained high across morning, afternoon, and night shifts, with minor fluctuations between shifts [19].

An American study also examined the perceptions of professionals and family members in a NICU and reported similar results to those identified in this research, with 71% of participants noting an environment with excessively high noise levels [47]. A German study found that despite recognizing the disruptive nature of the environment due to noise, there were challenges in identifying the specific types of sounds that were most bothersome [14].

In this study, noise was perceived as an “unpleasant” factor. The discomfort caused by noise in the work environment has negative impacts on professionals’ performance [10,38,51]. It is crucial to raise professionals’ awareness about their contribution to noise production when implementing strategies for noise control (Disher et al., 2017; Ahamed et al., 2018; Barsam, Barbosa, et al., 2019).

In the NICU environment, professionals experience a constant state of alertness due to various factors such as the unit’s physical structure, the provision of critical patient care, and the unique dynamics of each sector [52]. Therefore, it is not surprising that our participants reported the unit environment as stressful. The perception of stress can vary among individuals, depending on their ability to adapt to the challenges they face. Hence, what may be considered a stressful situation for one person may not be perceived as such by another [53]. Often, healthcare professionals become so immersed in their work that they may take a long time to recognize and address their own difficulties in coping with constant situations of discomfort [54]. Moreover, the inability to effectively respond to persistent stress can lead to chronic stress, which may ultimately contribute to the development of Burnout Syndrome, a condition frequently reported among healthcare professionals [4,55,56,57].

The present study found a significant and positive association between the perception of “excessive noise” and the perception of a “stressful shift,” indicating that noise could be a relevant factor in increasing stress among health professionals. All health professionals in our study reported positive criteria for the development of Burnout Syndrome, which may be attributed to the effects of noise and its potential association with certain sample characteristics also reported in other studies [18,58,59,60]. The sample consisted predominantly of female professionals with dependent children [18,58,59,60]. Another point to highlight is that the majority of participants had less than 5 years of experience in their respective roles, indicating that they were young professionals in the field. Another point to highlight is that the majority of participants had less than 5 years of experience in their respective roles, indicating that they were young professionals in the field [21,61], and the majority were nurses (70.7%) who have direct and close contact with patients and their families due to their professional responsibilities [57]. It is worth noting that nursing professionals have been particularly affected by situations that contribute to chronic stress, including personal factors, long working hours, interpersonal relationships [57], insufficient resources, inadequate reward systems, challenges in effective communication with superiors and other professionals [62], work overload, and patient interactions [53,63,64].

However, contrary to the existing literature that suggests a relationship between high noise levels and the occurrence of stress [65] and Burnout Syndrome [46], this study did not find a significant association between the occurrence of Burnout Syndrome and noise levels.

Nevertheless, it is concerning that all health professionals in our study reported positive scores for the presence of Burnout Syndrome, which aligns with the findings by Rahmati (2019) who identified psychological effects, anxiety, depression, and chronic stress as consequences of prolonged stress exposure [66]. Institutional policy can be an aggravating factor and contribute to chronic stress [67].

Considering the importance of appropriate levels of environmental noise for psychological, brain, and cardiovascular health [68], it is crucial to implement preventive and supportive measures to alleviate these symptoms and promote optimal occupational health, as well as enhance quality of life and work.

In our study, no significant changes were observed in cortisol levels between the beginning and end of the work shift in our sample, which suggests that stress in the work environment may not have had a direct impact on cortisol levels. This finding is consistent with a study conducted by Pérez-Valdecantos et al. (2021), where cortisol levels were evaluated during the work shift and rest time in professionals working in emergency care. The authors found that despite the absence of significant changes in cortisol levels, professionals still experienced high levels of stress during the work period. However, it is important to note that high levels of stress can sometimes enhance professional performance and decision-making and may not always be detrimental to the individual [69].

The fact that we did not observe significant changes in cortisol levels and in the incidence of noise-related Burnout may be associated with the fact that measurements and noise were high and with minimal variations between the sites evaluated, not allowing a comparison between professionals who worked in units with adequate noise levels. Initially, the hypothesis was that we would find significant differences between the units regarding the noise measurements that would allow this comparison, which did not occur. Another issue to consider is that these results may be associated with an adaptation of individuals to this adverse environment, as a possible coping strategy experienced by them [70]. In this study, the observed changes in salivary cortisol levels of health professionals, as well as their perceptions of tiredness at the end of the shift, “excessive” noise, and “stressful shift,” were not sufficient to differentiate the intensity of noise exposure to which the professionals were subjected. This could be attributed to the high and above-recommended noise levels in the environment, which prevented the analysis of differentiation with a group not exposed to noise. As a result, it was not possible to measure the impact of noise on participants’ stress levels.

Other studies have reported changes in cortisol levels among healthcare professionals working in different shifts. Niu et al. (2015) found reduced cortisol levels in nurses working the night shift compared to those working the day shift, and these night shift nurses took longer (approximately 4 days) to return to normal cortisol values, indicating a higher susceptibility to the effects of altered cortisol [15]. Similarly, Lin et al. (2022) found that nurses working the night shift had elevated stress levels, lower cortisol levels upon waking up, and reduced ability to perform basic daily tasks compared to nurses working the day and afternoon shifts [71].

Anjum et al. (2011) reported an increase in cortisol levels at the end of the day and low values in the morning in night workers, which is opposite to the hormone’s circadian variation [72]. Our study also observed a similar result among professionals working the night shift. We found that these individuals had elevated cortisol levels at the beginning of their shift, which normalized by the end of the shift. This particular finding warrants further evaluation in future studies to gain a deeper understanding of the underlying factors contributing to this pattern. These changes in cortisol have been associated with fatigue, burnout, exhaustion, and disruption of the hypothalamic–pituitary axis, which can contribute to physical and mental health problems [73,74,75].

It was observed that most professionals with normal cortisol levels were married. This finding aligns with a 2017 American study that investigated the stress of singlehood and found a positive correlation between being single and perceived stress [76]. Another study by Chin et al. (2017), which measured cortisol levels in men and women aged 21 to 55 years, reported higher cortisol values in singles compared to married individuals, suggesting that married people may be less susceptible to stress [77].

Consequently, it is important for health professionals to be aware of the importance of sleep, regular physical exercise, healthy nutrition, and coping strategies to effectively manage daily stressful situations [78].

The fact that all the evaluated environments had high noise levels may have been a limiting factor to the study, since it did not allow us to carry out a comparative study with environments with adequate noise levels.

The self-reported nature of the participants’ perception of Burnout symptoms is a limitation that needs to be addressed. However, the collection of salivary cortisol brought some strength to our study. It is worth mentioning that this study was carried out in NICU, Brazilian public university hospitals, which may introduce a selection bias. In general, these units constantly exceed their service capacity, being overcrowded, which impacts on noise levels, making the work more exhausting and highly demanding for the professional, which may have impacted the results of Burnout. The environments evaluated reflect the reality of Brazilian public hospitals, and the results of this research can be applied in units with the same institutional policy. However, for units that work with a fixed number of beds and that do not exceed their service capacity, they may present different results.

## 5. Conclusions

All the investigated NICUs exhibited high noise levels, surpassing the recommended standards set by the law. This indicates that the NICU environment does not meet the recommended criteria for acoustic comfort.

Although our health professionals reported the occurrence of Burnout Syndrome, it was not possible to establish a significant correlation with high noise levels due to the absence of a group exposed to recommended noise levels, which constitutes a limitation of the study. However, the participants’ perception of an excessively noisy environment was significantly associated with considering their work shift as stressful. This finding serves as an important indicator for the need to make changes in the noise levels. Addressing this issue is crucial to potentially reduce stress among health professionals and, consequently, mitigate the incidence and progression of Burnout Syndrome symptoms.

Therefore, it is imperative to implement strategies aimed at reducing noise levels in the evaluated units to ensure the safety and well-being of both professionals and patients. Further studies that include units with appropriate noise levels would be beneficial in assessing the influence and impact of noise on the occurrence of occupational stress among professionals and its effects on patients. Future research should take into account supplementary factors such as workload, support systems, or coping mechanisms to achieve a more all-encompassing comprehension of the correlation between noise and Burnout Syndrome. It would be beneficial to inform health policymakers and hospital managers that preventive measures are necessary to make the health system stronger and enhance the health of professionals.

## Figures and Tables

**Figure 1 healthcare-11-02002-f001:**
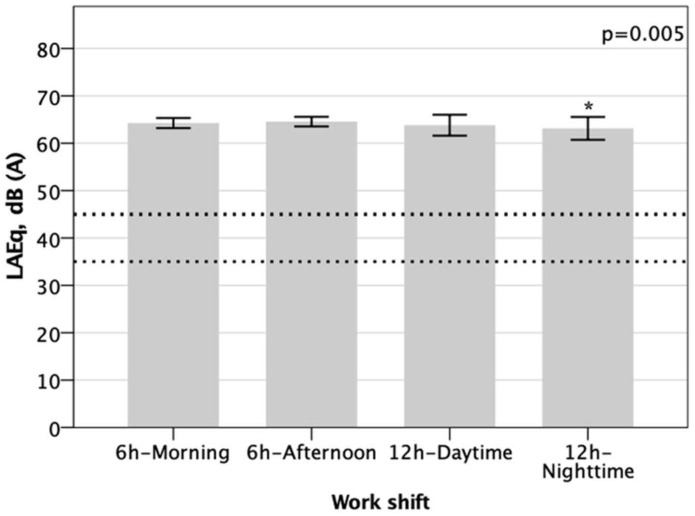
Comparison of the means of sound pressure levels according to health professionals’ work shifts. * *p* < 0.05.

**Figure 2 healthcare-11-02002-f002:**
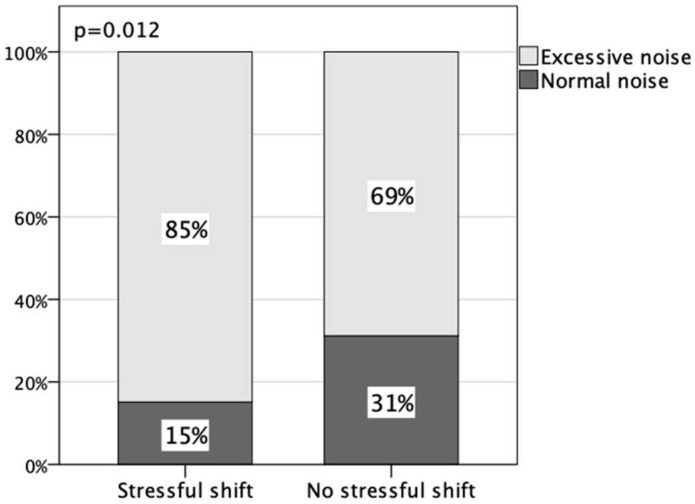
Relationship between the perception of excessive noise and the perception of stressful work shifts by health professionals. The chi-square test was applied.

**Table 1 healthcare-11-02002-t001:** Noise levels recommended for hospital environments.

Period	American Academy of Pediatrics (1997) (dBA)	World Health Organization (2006) (dBA)	Brazilian Association of Technical Procedures (2017) (dBA)
Day	45	35	-
Night	35	30	-
Range	-	-	35 to 40

**Table 2 healthcare-11-02002-t002:** Work environment, salivary cortisol, and subjective stress of health professionals.

Characteristics	*n* (%)
Considering the shift tiring (*n* = 226)	
Yes	128 (56.6)
No	98 (43.3)
Considering the shift stressful (*n* = 219)	
Yes	100 (45.7)
No	119 (54.3)
Considering noise level in the environment to be excessive (*n* = 204)	
Yes	160 (78.4)
No	44 (21.6)
Cortisol level (beginning of the shift) (*n* = 225)	
Normal	181 (80.4)
Increased	44 (19.6)
Cortisol level (end of the shift) (*n* = 230)	
Normal	222 (96.5)
Increased	8 (3.5)
Burnout occurrence (*n* = 246)	
Possibility	64 (26.1)
Initial	145 (58.9)
Installed	37 (15.0)

Categorical data expressed as absolute counts and percentages in parentheses. Only valid data were considered.

**Table 3 healthcare-11-02002-t003:** Relationship of sociodemographic characteristics and salivary cortisol at the beginning of the work shift of health professionals at NICU.

Characteristics	Cortisol Level at the Beginning of the Work Shift		
	Normal (*n* = 182)	Increased (*n* = 44)	*p*
Age	40 ± 10	37 ± 9	0.087
Gender			0.080
Male	12 (6.6)	0 (0)	
Female	170 (93.4)	44 (100)	
Marital status			0.010
Married	99 (55)	18 (41.9)	
Stable unit	10 (5.6)	9 (20.9)	
Single	56 (31.1)	14 (32.6)	
Divorced	15 (8.3)	2 (4.7)	
Having children			0.516
Yes	114 (63.3)	31 (72.1)	
No	61 (36.7)	12 (27.9)	
Function			0.213
Doctor	44 (24.4)	6 (14)	
Nurse	33 (18.3)	10 (23.3)	
Nurse technician	89 (49.4)	26 (60.5)	
Physiotherapist or speech therapist	14 (7.8)	1 (2.3)	
Time spent commuting (min)			0.846
<30 min	76 (41.7)	15 (34.9)	
31–60 min	68 (37.8)	20 (46.5)	
61 or more	36 (20)	8 (18.6)	
Having another job			0.277
Yes	72 (40)	13 (31)	
No	108 (60)	29 (69)	
Coming to work from another job			0.020
Yes	24 (13.3)	12 (27.9)	
No	156 (86.7)	31 (72.1)	
Continuous hours of work			0.316
6 h	3 (13.6)	2 (20)	
12 h	8 (36.4)	2 (20)	
18 h	3 (13.6)	4 (40)	
24 h	8 (36.4)	2 (20)	

Categorical data expressed as absolute counts and percentages in parentheses. Only valid data were considered.

**Table 4 healthcare-11-02002-t004:** Environmental noise levels of the NICUs and stress and work conditions of health professionals (*n* = 256).

Characteristics	L_Aeq_		*p*
	*n* (%)	Mean ± DP	
BURNOUT classification (*n* = 246)			0.391 ¹
Possibility	64 (26.1)	63.65 ± 2.26	
Initial	145 (58.9)	63.8 ± 1.69	
Frequent	37 (15)	63.26 ± 3.3	
Cortisol level (beginning of the shift) (*n* = 225)			0.062 ²
Normal	181(80.4)	63.87 ± 1.89	
Altered	44(19.6)	62.92 ± 3.18	
Cortisol level (end of the shift) (*n* = 230)			0.580 ²
Normal	222 (96.6)	63.67 ± 2.23	
Altered	8 (3.4)	64.11 ± 2.02	
Job function (*n* = 251)			0.073 ¹
Doctors	57 (22.7)	64.13 ± 1.04	
Nurse	48 (19.1)	63.09 ± 3.03	
Nursing technician	130 (51.8)	63.65 ± 2.13	
Physiotherapist and speech therapist	16 (6.4)	64.13 ± 1.45	
Work shift (*n* = 256)			**0.005** ^1,^*
6 h—Morning	34 (13.3)	64.24 ± 1.08	
6 h—Afternoon	23 (9.0)	64.56 ± 1.02	
12 h—Daytime	108 (42.2)	63.82 ± 2.21	
12 h—Night-time	91 (35.5)	63.14 ± 2.39	
Physical health (initial) (*n* = 243)			0.360 ¹
Very tired	25 (10.2)	63.76 ± 1.18	
Slightly tired	130 (53.5)	63.89 ± 1.84	
Rested	88 (36.3)	63.5 ± 2.35	
Physical health (Final) (*n* = 227)			0.558 ¹
Very tired	81 (35.7)	63.88 ± 2	
Slightly tired	103 (45.4)	63.56 ± 2.57	
Rested	43 (18.9)	63.52 ± 1.69	
Tiring shift (*n* = 226)			0.216 ²
Yes	128 (56.6)	63.83 ± 2.13	
No	98 (43.3)	63.49 ± 1.91	
Stressful shift (*n* = 219)			0.270 ²
Yes	100 (45.7)	63.54 ± 2.61	
No	119 (54.3)	63.85 ± 1.36	
Environment with excessive noise (*n* = 204)			0.603 ²
Yes	160 (78.4)	63.65 ± 2.42	
No	44 (21.6)	63.85 ± 1.47	

Quantitative data expressed as mean ± standard deviation. Bold value was regarded as significant (*p* < 0.05). Applied ANOVA test ¹; Student’s *t*-test ²; * Turkey test: *p* < 0.05 in “afternoon” vs “night-time”.

**Table 5 healthcare-11-02002-t005:** Correlation between cortisol levels with the noise levels that health professionals were exposed to during the work period, according to gender and to Burnout rating.

Characteristics	L_Aeq_		*p*
	*n*	rho	
Total group			
Cortisol level (end of the shift)	231	−0.054	0.417
Cortisol variation until the end of the shift (End–Start)	221	−0.001	0.983
Male			
Cortisol level (end of the shift)	12	−0.261	0.412
Cortisol variation until the end of the shift (End–Start)	12	0.062	0.849
Female			
Cortisol level (end of the shift)	219	−0.039	0.561
Cortisol variation until the end of the shift (End–Start)	209	−0.003	0.960
Burnout: “Possible”			
Cortisol level (end of the shift)	59	−0.064	0.632
Cortisol variation until the end of the shift (End–Start)	56	−0.05	0.713
Burnout: “Initial”			
Cortisol level (end of the shift)	133	−0.086	0.325
Cortisol variation until the end of the shift (End–Start)	127	−0.035	0.699
Burnout: “Frequent”			
Cortisol level (end of the shift)	33	−0.031	0.864
Cortisol variation until the end of the shift (End–Start)	32	0.032	0.861

Applied Spearman correlation coefficient, *p* < 0.05.

## Data Availability

The authors confirm that the data supporting the findings of this study are available within the article.

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
