# Peer review of "Excessive Noise in Neonatal Units and the Occupational Stress Experienced by Healthcare Professionals: An Assessment of Burnout and Measurement of Cortisol Levels"

_healthcare, 2023, doi:10.3390/healthcare11142002_

Round 1

Reviewer 1 Report

Introduction:

It is advisable that in the introduction of the original article you state the current situation on the subject under investigation (what is known and what is not), you have addressed them. However, I do not think I have found the justification, the knowledge gap to be filled. In addition, in the introduction section you should end by name the objective of the study (both the main and the specific objective) and the hypothesis of the study if there is one (I think it is missing in your introduction, justification and objective).

The bibliography consulted will be the one that tells us what is known and what is unknown about the phenomenon we are studying, for this reason it is recommended that you do not use articles older than five years. Could you look for articles that say more or less the same, but more up to date?

Try to update the citations: 1, 8, 14, 15 and 16.

Material and method.

You say that the participants personally answered a questionnaire that assessed socio-demographic information, working conditions, anthropometric indicators and self-perception of fatigue and stress.

What type of questionnaire was used for these questions? It would be advisable to add this in the article.

Author Response

Comment 1. Introduction: It is advisable that in the introduction of the original article you state the current situation on the subject under investigation (what is known and what is not), you have addressed them. However, I do not think I have found the justification, the knowledge gap to be filled. In addition, in the introduction section you should end by name the objective of the study (both the main and the specific objective) and the hypothesis of the study if there is one (I think it is missing in your introduction, justification and objective).

Response 1: The corrections were made.

Comment 2. The bibliography consulted will be the one that tells us what is known and what is unknown about the phenomenon we are studying, for this reason it is recommended that you do not use articles older than five years. Could you look for articles that say more or less the same, but more up to date? Try to update the citations: 1, 8, 14, 15 and 16.

Response 2: Done in part.  We improved some references.

Comment 3. Material and method. You say that the participants personally answered a questionnaire that assessed socio-demographic information, working conditions, anthropometric indicators and self-perception of fatigue and stress. What type of questionnaire was used for these questions? It would be advisable to add this in the article.

Response 3: The questionnaire was explained in more detail.

Reviewer 2 Report

Dear authors,

The study examines the association between noise levels in neonatal intensive care units (NICUs) and the occurrence of stress and Burnout Syndrome among healthcare professionals. While the research addresses an important topic, there are several areas that require attention and improvement to enhance the study's rigor and credibility.

Methodological Considerations: The study utilizes a descriptive, observational, and cross-sectional design. However, it would be beneficial to provide more details regarding the selection criteria for the hospitals and participants. Explaining how the participants were recruited and whether any specific inclusion or exclusion criteria were applied would improve the transparency of the study.

Please state how informed consent was obtained> written form? and where are stored?

Measurement of Noise Levels: The study measures sound pressure levels in the NICUs. Although the study mentions that the average sound pressure exceeded recommended levels, it would be helpful to discuss the specific guidelines or standards used for comparison. Providing references to established guidelines would strengthen the study's findings and allow for a better interpretation of the results.

Statistical Analysis and Associations: The study reports that there was no significant association found between noise levels and Burnout Syndrome, as well as no significant changes in cortisol levels. However, it is important to provide a thorough discussion on potential reasons for these findings. Factors such as sample size, characteristics of the participants, or other confounding variables should be considered and discussed to provide a comprehensive analysis of the results.

. It is crucial to acknowledge the limitations of the study in terms of generalizability. Discussing the potential implications of the study's findings on a broader scale and addressing any limitations that might impact the external validity would enhance the study's applicability.

Future Directions: 

Future research could consider additional variables such as workload, support systems, or coping strategies to provide a more comprehensive understanding of the relationship between noise and Burnout Syndrome.  It will be useful to state how politics or hospital management can be responsible for burnout in the lack of the preventive measures explained here>  https://www.mdpi.com/2227-9032/9/11/1550

minor revision

Author Response

Comment 1. Dear authors,

The study examines the association between noise levels in neonatal intensive care units (NICUs) and the occurrence of stress and Burnout Syndrome among healthcare professionals. While the research addresses an important topic, there are several areas that require attention and improvement to enhance the study's rigor and credibility.

Methodological Considerations: The study utilizes a descriptive, observational, and cross-sectional design. However, it would be beneficial to provide more details regarding the selection criteria for the hospitals and participants. Explaining how the participants were recruited and whether any specific inclusion or exclusion criteria were applied would improve the transparency of the study.

Response 1. The changes were made accordingly.

Comment 2. Please state how informed consent was obtained written form? and where are stored?

Response 2. The changes were made in the manuscript.

Comment 3. Measurement of Noise Levels: The study measures sound pressure levels in the NICUs. Although the study mentions that the average sound pressure exceeded recommended levels, it would be helpful to discuss the specific guidelines or standards used for comparison. Providing references to established guidelines would strengthen the study's findings and allow for a better interpretation of the results.

Response 3: Please, have a look at the Table 1. Furthermore, we added a sentence in the Methodology of the noise measurements (Page 5, Lines 190-191).

Comment 4. Statistical Analysis and Associations: The study reports that there was no significant association found between noise levels and Burnout Syndrome, as well as no significant changes in cortisol levels. However, it is important to provide a thorough discussion on potential reasons for these findings. Factors such as sample size, characteristics of the participants, or other confounding variables should be considered and discussed to provide a comprehensive analysis of the results.

Response 4: The changes were made accordingly.

Comment 5. It is crucial to acknowledge the limitations of the study in terms of generalizability. Discussing the potential implications of the study's findings on a broader scale and addressing any limitations that might impact the external validity would enhance the study's applicability.

Response 5: The changes were made.

Comment 6. Future Directions: Future research could consider additional variables such as workload, support systems, or coping strategies to provide a more comprehensive understanding of the relationship between noise and Burnout Syndrome.  It will be useful to state how politics or hospital management can be responsible for burnout in the lack of the preventive measures explained here>  https://www.mdpi.com/2227-9032/9/11/1550

Response 6: The changes were made in Conclusion.
